# Geographic and Specialty-Specific Disparities in Physicians’ Legal Compliance: A National-Scale Assessment of Romanian Medical Practice

**DOI:** 10.3390/healthcare11040499

**Published:** 2023-02-08

**Authors:** Codrut Andrei Nanu, Maria Cristina Plaiasu, Antoine Edu

**Affiliations:** 1Department No. 14 of Orthopedics, Anesthesia and Intensive Care, University of Medicine and Pharmacy “Carol Davila” Bucharest, 37 Dionisie Lupu Street, 020021 Bucharest, Romania; 2Doctoral School, University of Medicine and Pharmacy of Craiova, 2 Petru Rares St, 200349 Craiova, Romania; 3Department No. 14 of Obstetrics and Gynecology, University of Medicine and Pharmacy “Carol Davila” Bucharest, 37 Dionisie Lupu Street, 020021 Bucharest, Romania

**Keywords:** legal compliance, Romanian physicians, general practitioners, malpractice

## Abstract

Background: Physicians must respect their patients’ rights to informed consent, privacy, access to medical records, non-discrimination, treatment by a qualified doctor, and a second medical opinion. Compliance with patients’ rights is mandatory, and legal breaches are considered medical malpractice under Romanian law. This is the first study to assess physicians’ practices nationally and create a geographical map of legal compliance. Results: We examined survey responses of 2978 physicians, including 1587 general practitioners and 1391 attending physicians from high-risk specialties. According to the findings, 46.67% of physicians’ practices adhered to the law. Physicians’ practices were homogenous across the country’s regions. General practitioners were significantly more legally compliant than attending physicians were. Additionally, 94.02% of the physicians acknowledged malpractice anxiety, whereas only 17.67% had been accused of malpractice. Conclusions: Our findings emphasize the need for further research and to voice issues about Romanian physicians’ low level of legal compliance. This study provides a starting point for future studies to evaluate the benefits of interventional strategies in this field. Healthcare facilities should provide physicians with easily available resources when they are unsure about their legal obligations, and establish an observer organization that can detect unlawful conduct. Interventions should concentrate on education programs and expert guidance.

## 1. Introduction

The law plays an essential role in regulating the medical profession [1]. The law protects patients’ rights and requires physicians to comply with its provisions. Failure to meet the legal criteria may result in civil, disciplinary, and criminal liability for physicians.

The regulatory framework regarding the liability of healthcare professionals in Romania is governed by several laws and regulations, including Law no. 95/2006 regarding healthcare reform and the Law no. 46/2003 regarding patients’ rights [2]. These laws define the responsibilities of healthcare professionals and the circumstances under which they may be held responsible for any injuries they cause to patients. Furthermore, the Code of Ethics for Physicians defines the appropriate standards of professional conduct [3]. Additionally, the Romanian Criminal Code includes provisions related to medical malpractice and physicians can be prosecuted for criminal offences, including involuntary manslaughter, medical negligence, and confidentiality breaches. Overall, Romania’s regulatory framework is designed to hold physicians accountable for any harm caused to patients and compensate patients for damages.

The main areas of focus for Romanian medical law include informed consent, confidentiality, access to medical data, non-discrimination, working outside the scope of practice, and second medical opinion. Informed consent requires a shared decision-making process [4,5] found to be desirable by both physicians [6,7,8] and patients [9]. A physician’s legal responsibilities include evaluating a patient’s capacity to consent and identifying the legally authorized representatives of patients who are legally or physically incapable of consenting. Only adults and discerning people can validly express their agreement. All adult patients are presumed by law to have full decisional capacity. The lack of discernment of an adult patient can only be determined by the Forensic Psychiatric Expert Committee. According to Romanian legislation, only in one situation can physicians ask for consent from an underage patient. The exception pertains to patients under the age of 16 who are capable of making independent decisions regarding sexual and reproductive issues.

After an initial evaluation, physicians are required to inform and obtain a patient’s written consent. In the event of an underage patient, written agreement is requested from that patient’s parent or legal representative or, if that is not possible, from the closest relative. The closest relative refers to ascendants and significant relatives accompanying an underage patient. The law permits representation by relatives up to and including the fourth degree. In the case of a significantly incompetent patient, as determined by the mental forensic expertise committee, written agreement will be requested from the court-appointed legal representative. In the scenario of a legally competent adult patient with whom the medical practitioner cannot effectively communicate due to the patient’s medical condition at the time when consent is required, written consent might be acquired from the patient’s nearest spouse or closest relative. The term “closest relative” refers to parents, descendants, and relatives in the collateral line, up to the fourth degree, inclusive.

Additionally, Romanian law establishes mandatory standards for the disclosure of information process as well as a patient’s right to refuse disclosure [10]. In addition, the law specifies the minimum required information, which includes diagnosis, the nature and purpose of the medical act, a viable treatment alternative, the risks and consequences of treatment alternatives, the potential risks and consequences of treatment, the prognosis of the disease without treatment, the identity and professional status of health service providers, the rules and customs that patients must follow while hospitalized, and available medical services and how to use them. Medical intervention without consent is permitted under three legal conditions: an emergency, an unconscious or legally incapacitated patient, and an absence of legal representatives. A written report should be filled in by doctors after such medical interventions, describing the circumstances that enabled them to intervene without a patient’s consent.

The law protects patients’ confidentiality against all third parties, including family members, acquaintances, heirs, and other medical personnel uninvolved in a patient’s treatment. In general, a patient’s written agreement is mandatory for disclosing their medical data, with three exceptions: when data are required by another healthcare provider, in judicial proceedings, or when a patient poses a threat to themselves or others. Although third parties are denied medical information without a patient’s express consent, a patient has full access to their data. To receive their medical records, a patient must submit a form in this instance. Healthcare facilities must then comply within 48 h with a patient’s written request for copies of their medical dossier.

According to the law, patients are entitled to be treated equally, without discrimination, and to receive a second medical opinion. Moreover, physicians are not allowed to take medical actions that do not fit their field of expertise, except in emergencies when a patient’s life is in imminent danger and no competent medical staff is available. More and more studies are being conducted in post-communist countries to assess physicians’ practices from a legal perspective [11,12]. Some specialties are more prone to malpractice claims [13,14,15] and their chance of recurrence over time [16]. According to various international research, most complaints are reported against general practitioners and attending physicians [17,18]. In Romania, general practitioners have a lower risk of being accused of malpractice compared to attending physicians [19]. The COVID-19 pandemic challenged legal compliance, particularly regarding informed consent, patient selection, and performing outside the scope of practice [20,21].

Although physicians’ legal compliance is an essential aspect of hospital policies and a requirement for accreditation, there has been little or no effort in the last ten years to integrate legal aspects into the physicians’ curriculum. Except for the University of Medicine and Pharmacy “Carol Davila”, Bucharest, the national compulsory curriculum does not teach medical students about their future legal responsibilities.

Our study aimed to evaluate Romanian physicians’ legal compliance on a national and regional scale to provide a basis for subsequent comparison studies and to pave the way for future research in this area. Our paper creates a national map of physicians’ practices, allowing further research to investigate how they have evolved and to determine factors implicated in future developments. This is the first national study of Romanian general practitioners and physicians from high-risk specialties that assessed physicians’ legal compliance. As far as we know, it is among few at the international level.

## 2. Materials and Methods

We performed a retrospective study of the data made available by Medright Experts Ltd., Bucharest, Romania. The dataset was collected between 2009 and 2013 as part of the Romanian Sectorial Operational Program Human Resources Development POSDRU 20735 initiatives, no. 201735, “Quality in health through training of family doctors” and no. 56573, “Doctors in quality medical services”, financed by the European Social Fund. The projects were implemented in Romania by Medright Experts Ltd., the Romanian College of Physicians, the Romanian College of Dentists’ Medicine, the University of Medicine and Pharmacy “Gr. T. Popa”, Iasi, and the GCG Development Consulting Group. The main goals of these initiatives were to identify educational needs, train physicians, and develop a network of local accredited trainers in medical law.

In this retrospective study, we used only data received from families and attending doctors and excluded data retrieved from dentists, with the permission of Medright Experts Ltd., the appointed primary beneficiary of the initiatives mentioned above. With support from the Romanian College of Physicians, official invitations to participate in the educational programs were sent to all attending physicians specialized in surgery (general surgery, thoracic surgery, cardiovascular surgery, otorhinolaryngology, ophthalmology, orthopedics, plastic and reconstructive surgery), obstetrics and gynecology, anesthesia and intensive care and general practitioners with the right to free practice.

In the target group, 2978 physicians responded to invitations that pertained to the educational program only. Furthermore, before the courses, each group of physicians was asked “in situ” if they wished to engage in a research study assessing the liabilities of medical practice. Participants expressed their consent to use the collected data for educational and research purposes. Questionnaires were used to assess educational gaps and needs. After receiving their informed consent, respondents were invited to anonymously fill out a questionnaire addressing their medical practice from a legal point of view, and were instructed to choose the best answer for their current practice. The questionnaires were completely anonymous, and no personal data were collected. Additionally, the retrospective study was approved by the Ethics Committee of the University of Medicine and Pharmacy of Craiova under the number 87/09.06.2021.

The survey was previously developed and validated and contained multiple-choice questions regarding medical law, patients’ rights, and self-assessment questions (Appendix A) [22]. To evaluate physicians’ legal compliance, we retained the answer that complied with the law as valid.

We used Microsoft Excel (Microsoft Corp., Redmond, WA, USA), together with the XLSTAT add-on for MS Excel (Addinsoft SARL, Paris, France) and IBM SPSS Statistics 20.0 (IBM Corporation, Armonk, NY, USA) to process the data.

## 3. Results

Almost 20% of attending physicians from high-risk specialties and 10% of general practitioners registered in Romania participated in the study. The survey answers of 2978 physicians, comprising 1587 general practitioners and 1391 attending physicians, were examined. These physicians completed and returned the questionnaires to the trainers. From the general practitioner’s group, 28 physicians did not offer any answers to the legal compliance questionnaire, but decided to answer the self-assessment part of the questionnaire. They were taken into consideration only for those results.

The research shows that 46.67% of Romanian medical practice in the selected fields was in accordance with the law. Physicians’ practices were homogenous when assessing regional variations (*p* > 0.05). General practitioners were more legally compliant than attending physicians were, with 47.89% (CI: 2.37) versus 45.30% (CI: 2.34) accurate responses (*p* < 0.0001), respectively. Upon comparing each specialty’s variations between regions, it was observed that attending physicians’ practices remained homogenous, but general practitioners’ compliance varied significantly (*p* = 0.003) (Table 1).

The assessment of physicians’ practices revealed three significant challenges: operating within their specialty expertise, respecting patient confidentiality and access to medical data, and obtaining informed consent. Physicians were least inclined to complete a medical report after intervening in case of an emergency and to practice within the limits of their curriculum expertise. Additionally, confidentiality and medical data access rules were disobeyed by most physicians. In contrast, most respondents were compliant with regulations regarding discrimination and a second medical opinion (Table 2).

Regional disparities in physicians’ legal compliance are available for further comparative research if intended (Appendix A).

When self-evaluating their practice, 61.30% of respondents declared that they were aware of the medical protocols implemented at the hospital level, and only 11.13% acknowledged they had committed medical acts that might be considered medical faults.

Almost all (94.02%) physicians considered malpractice to be a serious and present threat, whereas only 17.67% had been accused of malpractice. There were variations between physicians’ specialties (Table 3). Additionally, general practitioners from the Northeast were most targeted by patients’ complaints, whereas most accusations against attending physicians were filed in the South region.

## 4. Discussion

Respect for patients’ rights is fundamental to the physician-patient relationship, and breaches may result in malpractice accusations [23]. Our research found that physicians’ compliance with legislative criteria governing patients’ rights was inadequate and unevenly distributed across specialties and geographical regions. More than half of national medical practices in general medicine and high-risk specialties violated the law, exposing physicians to substantial risks. The conduct of general practitioners was more legally compliant than that of attending physicians. This correlated with an increase in malpractice claims against attending physicians. On the other hand, general practitioners were less likely to be accused by their patients [9,24].

Despite this poor outcome, most physicians claimed awareness of the medical unit’s protocols. Although legal compliance may have been influenced by other vectors related to management issues and organizational ethics [25], we believe that in this case, physicians’ confusion may have been due mainly to a lack of legal knowledge. Other studies also identified insufficient legal knowledge, especially in emerging countries [26,27,28]. The following factors may have contributed to low legal awareness. First, relatively new legislation regarding patients’ rights and physicians’ responsibilities was introduced between 2003 and 2006; this was the result of proper harmonization of the national legislation with EU law, as a condition for Romania to join the EU in 2007. Second, there have been numerous law amendments; for instance, law no. 95/2006 regarding medical reform received 48 emendations between 2006 and 2012. Finally, there has been a lack of medical law education for medical students and physicians.

Furthermore, the study uncovered significant weaknesses in medical practice. A breach of confidentiality is one of these liabilities. As patients’ fear of being exposed works as a barrier to seeking medical care [29], trust in healthcare professionals is a critical element of medical practice [30,31]. Our research found that breaking confidentiality laws were common practice despite its importance. This finding is consistent with medical practice in other countries, where breaches frequently occur despite a solid commitment to confidentiality [32,33,34]. The proportion of attending physicians who complied with the confidentiality rules was much lower among our respondents. Additionally, in the previously cited studies, confidentiality breaches were due to publicly exposing patients’ data to third parties unintentionally or to medical personnel not involved in treating the patient. In our research, confidentiality breaches were found to be intentional. Romanian physicians admitted intentionally disclosing confidential information to third parties due to their family relationship with the patient. We believe this attitude originated in a lack of legal knowledge or a misunderstanding of the concept of privacy. Results also indicated that only a third of patients have full access to their medical data.

In contrast, physicians complied with informed consent rules, similar to other findings [35]. Given the study’s limitations, we could not determine whether physicians supplied all relevant information to patients, but could determine whether they were aware of the legal duty to inform. Most respondents (94%) from all specialties stated that they notified patients before performing a risky procedure. We observed significant specialty-related differences in acquiring written consent. Attending physicians maintained a high level of legal compliance when asked about documenting patient consent. More than 80% of attending physicians asked for written consent, as opposed to 40% of general practitioners. The disparity among specialties may be ascribed to hospitals’ management decisions to adopt legal protocols for informed consent, as opposed to the general practitioners’ self-determined implementation of such protocols in their private practices.

Physicians respected some patients’ rights more than they respected others’. A minority of physicians, for example, requested informed consent for biological sample collection and documented intervention provided to an incapacitated patient. Furthermore, only a third of physicians limited their practice to their specialty curricula and acted beyond their competence only in emergencies. In contrast, physicians were the most legally compliant with the standards of nondiscrimination and facilitating a second medical opinion.

The relatively low percentage of malpractice claims was consistent with previous studies [36]. Our result indicated that 17.67% of physicians had faced a malpractice accusation. This result was comparable with another current study that revealed that 16.1% of responding Romanian physicians confirmed they had been accused of malpractice [19]. Finally, physicians were highly afraid of being accused of malpractice. Fear of malpractice did not correlate with the small number of malpractice claims. Nonetheless, this fear could be legitimate in light of the numerous legal breaches of their practices.

## 5. Conclusions

Our findings emphasize the need for further research and to voice issues about Romanian physicians’ low level of legal compliance. These results point to the need for education programs, expert guidance when physicians are unsure about their legal obligations and an observer organization that can detect unlawful conduct. Our findings could serve as a foundation for future studies that assess the benefits of interventional strategies in this field.

## Figures and Tables

**Table 1 healthcare-11-00499-t001:** Physicians’ legal compliance within regions and specialties (%).

Region	General Practitioners	Attending Physicians
Bucuresti Ilfov	49.14	44.89
Center	49.06	46.36
North East	44.46	44.97
North West	47.92	45.35
South	46.41	44.09
South East	49.45	46.23
South West	47.25	44.96
West	50.32	46.46

**Table 2 healthcare-11-00499-t002:** Physicians’ legal compliance at a national level (%).

Question/Correct Answer	General Practitioners	Attending Physicians
Confidentiality and Data Access		
Is information about the health condition of a fully capable patient disclosed to third parties?/Yes, with express patient’s permission.	43.97	29.91
Is information about a patient’s treatment disclosed to third parties?/No.	38.27	18.76
In a media-interested case, do you allow press access to the patient?/Yes, with the patient’s express permission.	47.82	48.02
Are medical data regarding investigations, diagnosis, and treatment fully disclosed to the patient?/Yes, always, completely.	34.81	34.08
Informed consent		
Human biological samples are collected and analyzed./After obtaining the patient’s informed consent.	19.23	22.07
You will perform a potential risky maneuver on the patient. Do you inform the patient about the risks?/Yes, always in detail.	93.97	95.61
Do you obtain a patient’s written consent for performing a risky maneuver?/Yes, always.	45.38	82.31
Patient cannot express his consent and his health requires immediately intervention. The patient informed consent form is replaced by/A physician’s written report later added to patient’s medical file.	18.59	23.08
Discrimination		
Between two patients with similar medical condition which one do you prioritize?/None.	68.33	59.96
Competency breach		
How do you proceed when a patient required medical intervention that is beyond your competency?/I offer treatment in case of an emergency.	43.65	20.85
Second medical opinion		
The patient requests a second opinion from a physician outside of the hospital unit. What do you do?/I assist in receiving a second medical opinion.	72.76	63.70

**Table 3 healthcare-11-00499-t003:** Physicians’ legal compliance at a national level (%).

Question	Answers	General Practitioners	Attending Physicians	Total
Are you aware of the medical unit’s internal protocols?	Yes No There is no regulation	63.25	59.18	61.30
		21.89	23.46	22.64
		14.86	17.36	16.05
In the past three years, have you performed any medical act that could be deemed malpractice?	Yes	8.75	13.71	11.13
	No	42.61	45.44	43.96
	I can’t tell	48.64	40.85	44.91
Has a patient filed a complaint against you alleging malpractice?	Yes	8.70	27.49	17.67
	No	91.30	72.51	82.33
Do you view patients’ accusations of medical malpractice as a serious and present threat?	Yes	93.95	94.09	94.02
	No	6.05	5.91	5.98

## Data Availability

All data mentioned in the manuscript are available from the corresponding author if requested.

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
