# Peer review of "Geographic and Specialty-Specific Disparities in Physicians’ Legal Compliance: A National-Scale Assessment of Romanian Medical Practice"

_healthcare, 2023, doi:10.3390/healthcare11040499_

Round 1

Reviewer 1 Report

The study results are relevant and innovative, but some aspects regarding the methods need to be clarified.

It is important to include a publicly accessible link to the country's medical law legislation, so that authors from other nations can better understand the study.

The authors mention: “We used in this retrospective study only the data received from the family and at tending doctors”. Was such data requested based on informed consent? As this is a retrospective study, in the absence of consent, did Medright Experts Ltd request prior consent from participating physicians? Was the data provided in such a way as to guarantee anonymity?

The researcher describes that “With support from the Romanian College of Physicians, official invitations to participate in the educational programs were sent to all attending physicians”. It is not clear if the participants knew that they would be participating in a research, it is understood from the text that they were invited to an educational program. Clarify.

Has the study been reviewed by an independent research ethics committee? Include analysis opinion number by CEP.

A more impersonal writing is suggested, the authors write the results in third person plural.

Reviewer 2 Report

I found the study to be very interesting and its results morally disturbing, and it seems to have a large and adequate sample size. The survey was well-designed. 

My only suggestion for improvement is to do a heavy revision to the the introduction for the sake of clarity and to draw the reader in. I had to read and reread the introduction a few times to understand the study's background, topic, and purpose. For example, the second paragraph that begins "By law..." suffers from lack of parallel structure. The list could be written more clearly. 

A minor point: I'm interested in why the "correct answers" on the survey are the correct answers. For example, what is the relevant law or ethical principle behind some of these questions? For example, are there any limitations or exceptions in Romania on "full disclosure"? 

Round 2

Reviewer 1 Report

Explain whether the invitation to participate in the educational program included the participant's consent to the research. This is not yet explicit in the text.

Author Response

Dear Academic Reviewer,

We appreciate your thoughtful consideration. We have added the subsequent paragraph. We trust that by doing so, we have clarified the research methodology for the audience.

“Two thousand nine hundred seventy-eight physicians in the target group responded to invitations that pertained to the educational program only. Furthermore, before the courses, each group of physicians was asked “in situ” if they wished to engage in a research study assessing the liabilities of medical practice. Participants expressed their consent to use the collected data for educational and research purposes.”

Maria Cristina Plaiasu